# Segmentation as A Plug-and-Play Capability for Frozen Multimodal LLMs

## Abstract

Integrating diverse visual capabilities into a unified model is a significant trend in Multimodal Large Language Models (MLLMs). Among these, the inclusion of segmentation poses a distinct set of challenges. To equip MLLMs with pixel-level segmentation abilities, prevailing methods require finetuning the model to produce specific outputs compatible with a mask decoder. This process typically alters the model's output space and compromises its intrinsic generalization, which undermines the goal of building a unified model. We introduce *LENS* (**L**everaging k**E**ypoi**N**ts for MLLMs' **S**egmentation), a novel plug-and-play solution. *LENS* attaches a lightweight, trainable head to a completely frozen MLLM. By refining the spatial cues embedded in attention maps, *LENS* extracts keypoints and describes them into point-wise features directly compatible with the mask decoder. Extensive experiments validate our approach: *LENS* achieves segmentation performance competitive with or superior to that of retraining-based methods. Crucially, it does so while fully preserving the MLLM's generalization capabilities, which are significantly degraded by finetuning approaches. As such, the attachable design of *LENS* establishes an efficient and powerful paradigm for extending MLLMs, paving the way for truly multi-talented, unified models.

## 1 Introduction

Built on Large Language Models (LLMs), Multimodal LLMs (MLLMs) have demonstrated generalized visual understanding, most notably through their ability to ground language instructions in specific image regions (Zhang et al., 2025). This property connects high-level semantics with visual space, paving the way to reformulate different vision tasks into a unified visual-instruction-controlling manner (Wu et al., 2024c; Lai et al., 2024; Ma et al., 2024). As this trend unfolds, MLLMs are expected to encompass a full spectrum of visual tasks, including recognition (Liu et al., 2023c), detection (Wu et al., 2024c), and even dense, pixel-level segmentation (Lai et al., 2024).

Yet, integrating segmentation capability presents a unique challenge, as its dense pixel-mask outputs cannot be natively expressed by the text-generative nature of LLMs, nor is there a large-scale segmentation corpus for autoregressive pre-training (Lai et al., 2024). This skill must instead be transferred from a conventional, pre-trained segmentation model (Lai et al., 2024; Qian et al., 2025; Wu et al., 2024b). As illustrated in Fig. 1a, prevailing approaches feed MLLM features into SAM's decoder (Ravi et al., 2024), which then maps them into masks. Notably, a significant mismatch exists: segmentation decoders are designed for low-level spatial cues (*e.g.*, points or boxes), whereas MLLMs produce high-level, abstract semantic features (Jiang et al., 2025). To bridge this gap, existing solutions always involve extensively fine-tuning the MLLM with both segmentation and generation objectives, thereby training it to produce features compatible with the segmentation decoder (Qian et al., 2025). Despite its straightforwardness, these approaches prove highly effective for instruction-controlled segmentation.

This effectiveness, however, comes at a cost. The dual-objective training introduces an inherent tension between model's capabilities: generative tasks thrive on abstract, sparse semantics, whereas segmentation requires direct, spatial features (Liu et al., 2024b). Although large models can accommodate both, this compatibility is fragile and often degrades other general-purpose abilities (Wu et al., 2024b). Take LISA (Lai et al., 2024) as an example, which is concurrently trained to generate a special `[SEG]` token and adapt its corresponding features to be compatible with the SAM-based

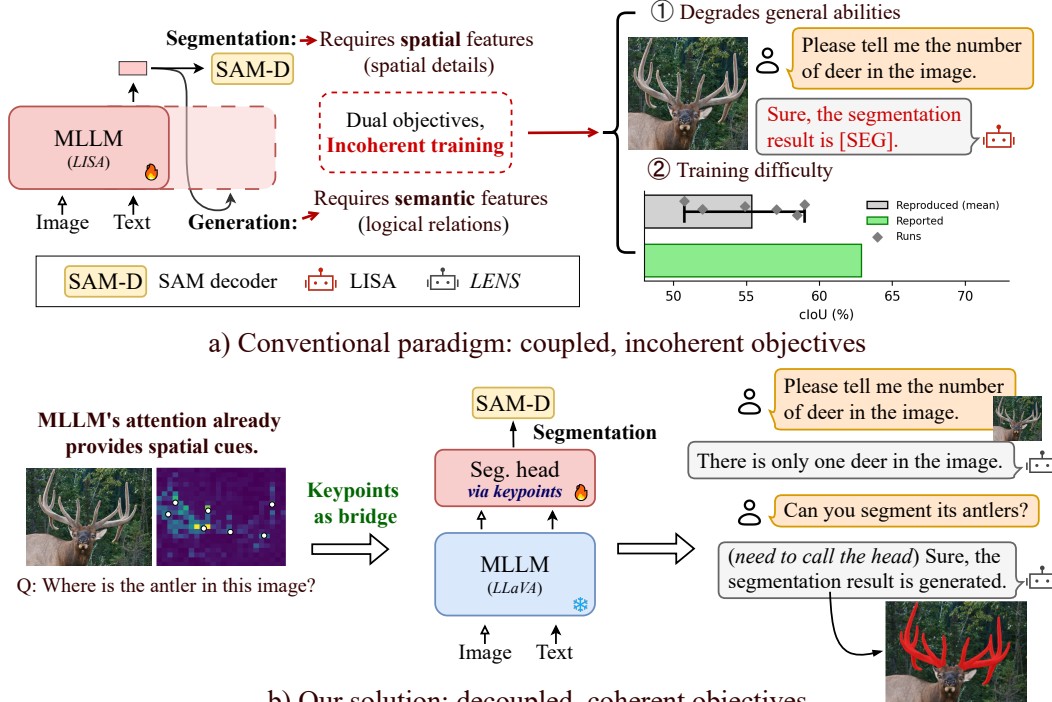

Figure 1: **Conventional architecture for MLLM segmentation vs. *LENS*. (a)** Conventional methods (*e.g.* LISA (Lai et al., 2024)) fine-tune an MLLM for both generation and segmentation tasks, leading to conflicting objectives that undermine the model's general capabilities and training stability. **(b)** *LENS* decouples these roles: a frozen MLLM is dedicated to reasoning, while a lightweight head is trained exclusively for segmentation. The head can be adaptively invoked by the model when needed, allowing the MLLM to serve as a unified vision model capable of handling diverse tasks.

decoder. Consequently, it frequently defaults to segmentation-focused responses like "Sure, the segmentation result is `[SEG]`", even for a completely unrelated counting query (*c.f.*, Fig. 1a). This narrow focus reduces the MLLM to a single-task tool, causing its performance on the general-purpose benchmark like MMBench (Liu et al., 2024c) to plummet to near-zero. Such an outcome fundamentally contradicts the goal of building unified and versatile vision models.

Another drawback, as noted by prior studies (Chen et al., 2024b; Zhu et al., 2025), is that combining segmentation and generation losses increases optimization complexity, which we also observe in our reproductions (*c.f.* Fig. 1a). Dual-objective models are highly sensitive to training configurations and require extensive hyperparameter tuning to achieve competitive results.

Motivated by these limitations, we argue that segmentation should be introduced as a **plug-and-play** capability, one that enhances the MLLM without compromising its foundational strengths. An intuitive strategy is to freeze the MLLM entirely and train an external head dedicated to converting its features for segmentation. However, this simple architectural change is insufficient. Frozen MLLMs provide only semantic features, having already discarded most of the fine-grained spatial details critical for segmentation (Jiang et al., 2025). This flaw requires more than a mere structural modification; it necessitates a paradigmatic shift in how MLLMs' features are leveraged.

Our approach sparks this shift through a crucial insight: an MLLM's internal attention mechanisms already provide the spatial cues (Jiang et al., 2025; Wang et al., 2025; Zhang et al., 2025). As illustrated in Fig. 1b, when an MLLM processes a query, a distinct attention pattern emerges over the image, with high-scoring regions corresponding to the object of interest (*e.g.*, the antler). This allows us to repurpose the segmentation head for a more direct task: refining these attention-derived spatial cues into keypoint coordinates and using the MLLM's semantic judgment to assign corresponding descriptions (labels). These **keypoint–description** pairs act as direct prompts for the SAM decoder, effectively bridging the MLLM's internal representations with the segmentation model's in-

put requirements. By leveraging the MLLM's native abilities for both localization via attention and verification via semantics, this process makes the head's training remarkably coherent.

We call this architecture *LENS* (**L**everaging k**E**ypoi**N**ts for MLLMs' **S**egmentation), which equips MLLMs with segmentation capability while keeping the backbone entirely frozen. This design avoids degrading the MLLM's general-purpose abilities and delivers substantial efficiency gains: Since the MLLM is used purely in inference mode, training costs are greatly reduced. Meanwhile, the segmentation head functions as a modular, plug-and-play tool that can be invoked on demand, enabling seamless integration into agent-based systems (*c.f.* Fig. 1b). Overall, our contributions can be summarized as follows:

1. We introduce *LENS*, a novel segmentation architecture to operate on a completely frozen MLLM backbone. This decoupled paradigm is designed to preserve the integrity of the MLLM's general-purpose abilities, thereby resolving a central flaw in prior fine-tuning methods.
2. We demonstrate how spatial cues from an MLLM's internal attention can be refined into SAM-compatible prompts, with keypoints serving as the bridge between high-level reasoning and pixel-level segmentation.
3. *LENS* achieves state-of-the-art performance on multiple segmentation benchmarks while notably reducing training costs, as the core MLLM is utilized purely for inference. Its efficiency and plug-and-play design offer a practical and scalable solution for unified vision models.

## 2 RELATED WORK

**Multimodal Large Language Models (MLLMs).** The advent of MLLMs represents a paradigm shift in computer vision, driven by the powerful reasoning capabilities inherited from their underlying LLMs (Kaplan et al., 2020; OpenAI, 2024; Google, 2025). Architectures like LLaVA (Liu et al., 2023c; 2024a), InstructBLIP (Dai et al., 2023), and Qwen-VL (Bai et al., 2023) typically connect a visual encoder to a pre-trained LLM core via lightweight, parameter-efficient modules. This architectural integration enables the generation of text grounded in visual input, establishing a robust and sophisticated alignment between language and vision. There are two primary ways to leverage this intrinsic vision-language spatial association: either the model directly articulates its understanding through generated text (Bai et al., 2025; Peng et al., 2023), or its internal mechanisms, such as attention, can be decoded to reveal its spatial cues (Zhang et al., 2025; Wang et al., 2025).

*Spatial Cues in Attention* Recent investigations have consistently shown that the attention mechanisms within MLLMs serve as a natural bridge between textual tokens and their corresponding image regions (Zhang et al., 2025; Wang et al., 2025; Yang et al., 2025; Yu et al., 2024; Kang et al., 2025). When conditioned on paired image-text input, attention maps highlight the regions most relevant to the textual description, effectively providing coarse spatial cues of the target (Wang et al., 2025). Crucially, this is not an idiosyncratic feature of any single architecture but a universal, emergent property observed across a diverse range of models (Wang et al., 2025; Zhang et al., 2025; Yu et al., 2024; Kang et al., 2025). This phenomenon arises organically from the model's objective to generate text that is contextually grounded in the visual input; to accurately describe an object, the model must first "look" at it. Consequently, attention maps offer a robust and direct source of spatial information, making them an ideal foundation for dense prediction tasks like segmentation, which demand more granular guidance than textual outputs can offer.

**Segmentation Models.** Early image segmentation paradigms, such as semantic and panoptic segmentation (Badrinarayanan et al., 2017; Long et al., 2015; Ronneberger et al., 2015), were predominantly closed-set, operating on a fixed vocabulary of object categories. A recent shift towards open-vocabulary segmentation has been driven by promptable models that accept diverse control signals (Ravi et al., 2024; Kirillov et al., 2023; Liu et al., 2023b; Wu et al., 2024a; Ren et al., 2024a; Zou et al., 2023a; Liu et al., 2023a; Zou et al., 2023b). These range from low-level spatial prompts (*e.g.*, points, boxes) in models like SAM (Kirillov et al., 2023) to explicit textual phrases in Referring Expression Segmentation (RES) (Hu et al., 2016; Liu et al., 2023b; Wu et al., 2024a; Ren et al., 2024a; Liang et al., 2023; Zou et al., 2023a). Despite their flexibility, these methods are fundamentally limited by their dependence on direct, literal prompts. They lack the higher-level reasoning ability needed to ground complex, inferential semantics in pixel space, which motivates the development of dedicated reasoning-based segmentation models.

*Reasoning Segmentation Models.* As an advanced form of RES, reasoning segmentation targets objects that are only implicitly referenced and must be inferred from descriptive cues (*e.g.*, segment "the organ used for defense" instead of just "antler"). The inherent demand for strong comprehension and reasoning has naturally positioned MLLMS as the foundational backbone for this task (Lai et al., 2024; Rasheed et al., 2024; Ren et al., 2024b; Wu et al., 2024b; Xia et al., 2024; Qian et al., 2025). LISA (Lai et al., 2024) pioneered this task by training an MLLM to emit a special token whose feature is then fed into a SAM-like decoder; the model is jointly optimized on large-scale mixtures of instruction-following and segmentation data to transfer the MLLM's reasoning ability to the segmentation domain. Building on this paradigm, SESAME (Wu et al., 2024b) introduces negative examples to enable refusal of non-segmentable queries, while READ (Qian et al., 2025) analyzes the underlying mechanism and proposes similarity-based objectives to further refine performance. Although viable, these methods all rely on heavy joint training. Even with optimizations like LoRA (Hu et al., 2022), tightly coupling the objectives for generation (semantics) and segmentation (spatial) creates a trade-off. This often leads to the MLLM becoming over-specialized, compromising its foundational general-purpose abilities.

## 3 PROPOSED *LENS*

In this section, we present *LENS*, a novel architecture that equips a frozen MLLM with segmentation in a plug-and-play manner. As illustrated in Fig. 2a, *LENS* consists of three stages: a lightweight head (§3.1), a keypoints extraction and description module (§3.2), and a mask decoder (§3.3). The central innovation of this design is the use of keypoints from the MLLM's internal attention maps as a bridge that intrinsically unifies the stages. We next detail each stage, followed by the training objectives and configurations (§3.4).

### 3.1 SEGMENTATION HEAD

The segmentation head receives semantic features from the MLLM, (i) refines the attention dependencies to increase target-region keypoints, and (ii) provides a decision on whether the attention-highlighted regions should be identified as segmentation targets.

**Architecturally**, the head is a two-layer transformer, mirroring a single MLLM layer for consistency. Its dual roles impose two requirements on input features: (i) strong cross-modal attention[1], and (ii) sufficient semantics to identify grounded targets. As shown in prior work (Zhang et al., 2025; Jiang et al., 2025), shallow layers are deficient in semantics, while deep layers exhibit diminished cross-modal attention. Thus, we adopt intermediate features (*e.g.*, the 14th layer in LLaVA-1.5-7B), which best balance these properties.

Given an input image $I$ and instruction $T$, we denote their intermediate features as $F_i \in \mathbb{R}^{L_i \times d}$ for the image (with $L_i = 576$ in LLaVA-1.5-7B) and $F_t \in \mathbb{R}^{L_t \times d}$ for the text. These are concatenated into $H_{\text{in}}^1 = [F_i; F_t] \in \mathbb{R}^{L \times d}$ which serves as the input to the head. For simplicity of exposition, we assume a fixed ordering where text features are always appended after the image features.

**Layer 1: Attention Refinement**. While MLLM attention maps can localize grounded targets, they are not tailored for segmentation. As observed by Darcet et al. (2024), they often highlight contextual regions useful for text generation but irrelevant to segmentation. Therefore, the first layer is to re-calculate and refine these maps, explicitly training them to suppress extraneous activations and selectively highlight only the regions corresponding to the intended target.

To achieve this, the layer first computes a full attention map $A^1$ over all input tokens. We then *aggregate* the attentions from text to image tokens by averaging their weights, yielding the text-to-image grounding map $A_c^1$ (Fig. 2b). The computation is as follows:

$$A^1 = \text{Softmax}\left(\frac{QK^\top}{\sqrt{d}} + M_a\right), \quad A_c^1 = \frac{1}{L_t} \sum_{k=L_i+1}^{L_i+L_t} A^1[k, 1:L_i], \tag{1}$$

where $Q$ and $K$ are linear projections of input features $H_{\text{in}}^1$, and $M_a$ is the causal attention mask. $A_c^1$ is explicitly optimized through training, and $A^1$ is used to produce the output features $H_{\text{out}}^1 \in \mathbb{R}^{L \times d}$.

---

[1]Attention from text to image tokens.

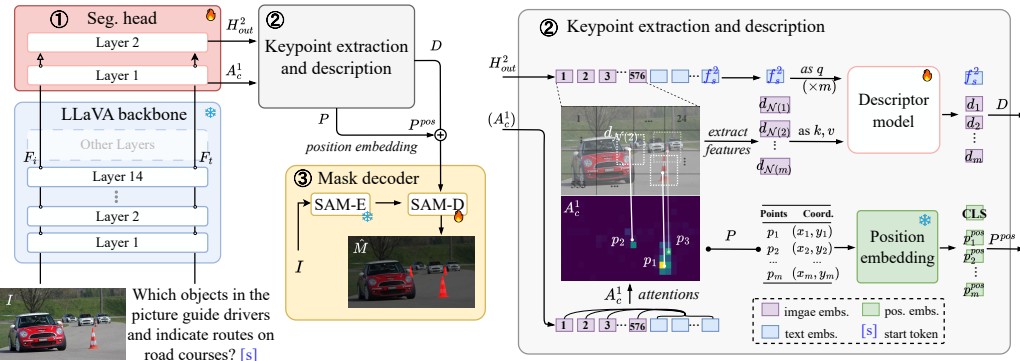

a) Overview of *LENS*       b) Zoom-in view

Figure 2: ***LENS* architecture. (a)** Overall architecture built on LLaVA-1.5-7B (Liu et al., 2024a), consisting of three stages: ① a segmentation head that refines attention and semantic features, ② keypoint extraction and description, and ③ a mask decoder that takes fused keypoint descriptions and coordinates to predict the final mask. Trainable and frozen modules are indicated, and the MLLM runs only in inference mode. **(b)** Zoom-in view of the attention and keypoint components.

**Layer 2: Feature Enhancement.** Since the attention in Layer 1 is explicitly optimized, the resulting representation $H_{\text{out}}^1$ may carry semantic bias. Layer 2 aims to mitigate this bias and enhance the discriminative semantics of $H_{\text{out}}^1$, thereby producing the output $H_{\text{out}}^2$. We expect the feature of the *start*-of-answer token[2] $f_s^2$ to align with the image features in $H_{\text{out}}^2$, enabling it to serve as a semantic query for identifying the image regions that correspond to segmentation targets.

Overall, the process of the segmentation head can be summarized formally as follows:

$$A^1, H_{\text{out}}^1 \ \leftarrow \ \text{Layer}_1([F_i; F_t]), \quad A_c^1 \ \leftarrow \ \text{Aggregate}(A^1), \quad H_{\text{out}}^2 \ \leftarrow \ \text{Layer}_2(H_{\text{out}}^1). \quad (2)$$

### 3.2 KEYPOINT EXTRACTION AND DESCRIPTION

The second stage extracts points from the high-value regions of $A_c^1$, which serve as indicators for segmentation. We define these as *keypoint* regions. Each keypoint is then *described* as positive if the semantics of its image feature in $H_{\text{out}}^2$ match $f_s^2$, and negative otherwise. The resulting positions and descriptions together form the prompts, which serve as the structured input to the SAM decoder (*c.f.* Fig. 2b).

**Keypoint Extraction.** The attention map $A_c^1$ is reshaped into a 2D heatmap, from which keypoints are extracted via Non-Maximum Suppression (NMS). Local maxima are selected as candidate positions, and up to $m$ keypoints are retained.

Since $A_c^1$ is defined at the patch level, the heatmap resolution is low and coordinates are confined to grid positions, which is suboptimal for pixel-level segmentation. To mitigate this, we apply a *sub-pixel* refinement[3] that shifts grid-aligned coordinates toward the underlying peak locations. The refined set is denoted $P = \{p_i\}_{i=1}^m$, where each $p_i = (x_i, y_i)$. These keypoints are then encoded into position embeddings $P^{pos}$ compatible with the SAM decoder. The implementation of this encoding is deferred to §3.3.

**Keypoint Description.** To determine whether each keypoint corresponds to a positive or negative region, we extract its associated semantic features. Specifically, at each coordinate we *sample* the image feature from $H_{\text{out}}^2$ via interpolation, and further sample from a $p \times p$ neighborhood to enrich the semantic representation. This yields a local feature set $d_{\mathcal{N}(i)}$ for each keypoint $p_i$.

We then leverage the global *start*-of-answer token feature $f_s^2$ as a query to determine whether the region of $p_i$ should be segmented. The neighborhood features $d_{\mathcal{N}(i)}$ serve as keys and values in a de-

---

[2] Derived from the question's final token $\texttt{[s]}$ in Fig. 2b.

[3] Implemented with a Newton–Raphson update; details in the supplementary material.

scriptor model[4], where *cross-attention* is performed to produce discriminative descriptions $\{d_i\}_{i=1}^m$. Through the interaction between $f_s^2$ and neighborhood features, these descriptions are expected to acquire the discriminative capacity needed for positive/negative interpretation.

**Global Description.** While each keypoint yields a local description, these remain independent and may contain redundancy or spatial overlap. To promote coherence among them, we further introduce $f_s^2$ as a global semantic descriptor within the descriptor model. Through a subsequent *self-attention* operation, $f_s^2$ interacts with all local descriptions $\{d_i\}_{i=1}^m$, enabling global context to regularize redundant or spatially overlapping instances while simultaneously consolidating information back into $f_s^2$. The final description set is defined as $D$.

The process of keypoint extraction and description can be summarized as:

$$P \;\leftarrow\; \text{Sub-pixel}\big(\text{NMS}(A_c^1)\big)\,, \qquad \{d_{\mathcal{N}(i)}\}_{i=1}^m \;\leftarrow\; \text{Sample}\big(\mathbf{F}_i^2, P\big)\,,$$
$$\{d_i\}_{i=1}^m \;\leftarrow\; \text{Cross-attn}\big(f_s^2, \{d_{\mathcal{N}(i)}\}_{i=1}^m\big)\,, \qquad D \;\leftarrow\; \text{Self-attn}\big(\{f_s^2\} \cup \{d_i\}_{i=1}^m\big)\,. \tag{3}$$

where both $\mathbf{F}_i^2$ (image tokens) and $f_s^2$ (start token) are taken from $\mathbf{H}_{\text{out}}^2$.

## 3.3 MASK DECODER

At this stage, we have the keypoint set $P \in \mathbb{R}^{m \times 2}$ and the description set $D \in \mathbb{R}^{(m+1) \times d_s}$, where $d_s$ matches the embedding dimension of the SAM decoder. The keypoints naturally match the point-based prompts of SAM, and the descriptors play the role of label embeddings. This structural alignment allows our outputs to be seamlessly integrated into the SAM decoder.

**Position Embedding.** We adopt SAM's point *position encoder* to transform the keypoints $P$ into embeddings $P^{\text{pos}} = \{p_i^{\text{pos}}\}_{i=1}^m$. Since the global descriptor $f_s^2$ lacks a spatial position, we introduce a learnable `[CLS]` embedding as its positional counterpart. This yields both the position embeddings $P^{\text{pos}}$ and the label embeddings $D$ required by the SAM decoder.

The summed embeddings are fed into the decoder to generate the final mask $\hat{M}$:

$$P^{\text{pos}} \;\leftarrow\; \{\text{CLS}\} \cup \text{PosEnc}(P), \qquad \hat{M} \;\leftarrow\; \text{Decoder}\big(D \oplus P^{\text{pos}}, F_{\text{img}}^{\text{SAM}}\big)\,, \tag{4}$$

where $\oplus$ denotes element-wise addition and $F_{\text{img}}^{\text{SAM}}$ are the image features from the SAM encoder.

## 3.4 TRAINING AND USAGE

**Training.** Our model is trained end-to-end with a composite loss function consisting of two components: an attention loss $\mathcal{L}_{\text{attn}}$ and a segmentation loss $\mathcal{L}_{\text{seg}}$.

*Attention Loss.* $\mathcal{L}_{\text{attn}}$ provides direct supervision for the cross-modal attention map $A_c^1 \in [0,1]^{h \times w}$. Given the ground-truth binary mask $M \in \{0,1\}^{h \times w}$, we use the binary cross-entropy (BCE) loss to enforce alignment between $A_c^1$ and $M$:

$$\mathcal{L}_{\text{attn}} = -\frac{1}{hw} \sum_{i=1}^h \sum_{j=1}^w \Big[ M_{i,j} \log A_{c,i,j}^1 + (1 - M_{i,j}) \log \big(1 - A_{c,i,j}^1\big) \Big]. \tag{5}$$

*Segmentation Loss.* For the segmentation loss $\mathcal{L}_{\text{seg}}$, we follow the practice of LISA (Lai et al., 2024) and adopt a combination of Dice loss and BCE loss applied to the final predicted mask $\hat{M} \in [0,1]^{h \times w}$. It's the weighted sum of the Dice and BCE losses:

$$\mathcal{L}_{\text{seg}} = \lambda_{\text{dice}} \mathcal{L}_{\text{dice}} + \lambda_{\text{bce}} \mathcal{L}_{\text{bce}}. \tag{6}$$

*Overall Objective.* The overall training objective combines the two losses:

$$\mathcal{L} = \mathcal{L}_{\text{seg}}(\hat{M}, M) + \mathcal{L}_{\text{attn}}(A_c^1, M). \tag{7}$$

**Usage.** Unlike token-based triggering mechanisms, *LENS* relies on the MLLM to determine through question answering whether segmentation should be activated. The routing can be implemented using tool frameworks (Chase, 2025) or thinking-based control (Liu et al., 2025). We center on *LENS*'s design; implementation details and illustrative demonstrations appear in the supplementary material.

---

[4]The detailed structure is described in the supplementary material.

# 4 EXPERIMENTS

## 4.1 EXPERIMENTAL SETUP

**Implementation Details.** For a fair comparison with prior works, we adopted the widely used LLaVA-1.5-7B (Liu et al., 2024a) as the backbone for the main experiments, while SAM is instantiated with ViT-H. We used the $14^{\text{th}}$ layer as the intermediate representation, set $m = 16$, and adopted a neighborhood size of $3 \times 3$. The optimizer was AdamW with a learning rate of $5 \times 10^{-5}$. The loss weights for Dice and BCE in $\mathcal{L}_{\text{seg}}$ were set to 2 and 4, respectively. Unless otherwise specified, other training settings followed LISA (Lai et al., 2024).

**Training Datasets.** Following the dataset organization in LISA, we considered three categories: (1) semantic segmentation datasets including ADE20K (Zhou et al., 2019), COCO-Stuff (Caesar et al., 2018), PACO-LVIS (Ramanathan et al., 2023), PASCAL-Part (Chen et al., 2014), and Mapillary Vistas (Neuhold et al., 2017); (2) referring segmentation datasets including RefCLEF, RefCOCO, RefCOCO+ (Kazemzadeh et al., 2014), and RefCOCOg (He et al., 2017); and (3) reasoning segmentation dataset ReasonSeg (Lai et al., 2024). Note that *LENS* was trained only with segmentation objectives and preserves general abilities without extra VQA data.

**Evaluations.** Our assessment proceeded from a *comprehensive* perspective to a *segmentation-specific* one. At the comprehensive level (*c.f*. Table 1), *LENS* excels in training efficiency while preserving general abilities (benchmark settings are provided in the supplementary material). At the segmentation level (*c.f*. Table 2 and Table 3), *LENS* establishes state-of-the-art results on reasoning segmentation and RES, measured by gIoU (per-image IoU) and cIoU (dataset-level IoU).

**Baselines.** We directly compared against methods that require fine-tuning MLLMs for segmentation, including LISA (Lai et al., 2024), SESAME (Wu et al., 2024b), and READ (Qian et al., 2025). In addition, following LISA, we also included traditional baselines for RES task for further comparison on segmentation tasks, as reported in Table 3.

## 4.2 COMPREHENSIVE EVALUATION

We compared model backbones, training cost, training data, and resulting segmentation and general abilities (*c.f*. Table 1). Training cost was measured under the DeepSpeed (Rasley et al., 2020) ZeRO-2 setting with 8 GPUs and a batch size of 2. *Seg* denotes the segmentation data (see §4.1); *FP-Seg* denotes an augmented version of *Seg*, constructed using FP-RefCOCO(+/g) (Wu et al., 2024b) and R-RefCOCO(+/g) (Wu et al., 2024a). *VQA* is the instruction corpus from LLaVA. For segmentation evaluation, we reported cIoU on ReasonSeg. For general capability evaluation, we adopted MME Fu et al. (2023), MMBench Liu et al. (2024c), MMMU (Yue et al., 2024), and MMStar (Chen et al., 2024a) benchmarks. Further details are provided in the supplementary material.

**Training Efficiency.** As shown in Table 1, *LENS* is highly efficient. Since the MLLM is used only for inference, gradients are not back-propagated through it, allowing distributed execution or even pre-caching. As a result, the MLLM itself requires as little as 16 GB of memory. Overall, *LENS* reduces training memory to one-third while still achieving the best comprehensive performance.

**Avoiding the Multi-Objective Trade-off.** *LENS* functions as a plug-and-play tool that the MLLM can invoke when needed (*e.g*., via chain-of-thought reasoning), without relying on special tokens or auxiliary objectives. It is trained exclusively on segmentation data and remains fully decoupled from the MLLM's generative learning. By contrast, prior approaches entangle segmentation with generation through additional tokens and losses, which drastically compromise general-purpose ability (MMBench accuracy drops from 66.5 to 0 for READ and LISA). By avoiding this trade-off, *LENS* preserves unified vision–language capability without incurring additional cost.

**SOTA Segmentation with Preserved General Capabilities** Our plug-and-play design endows *LENS* with state-of-the-art segmentation ability while retaining the general capabilities of the underlying MLLM. Compared to LISA, *LENS* achieves higher segmentation performance (57.3 *vs*. 56.9) without the collapse of general abilities (MMBench, MMMU, and MMStar all remain on par with LLaVA-1.5-7B, whereas LISA drops to zero). READ shows slightly better segmentation (58.6) but benefits from a stronger backbone and larger training data, while still suffering from degraded generality. SESAME attempts to balance segmentation and understanding through refined data en-

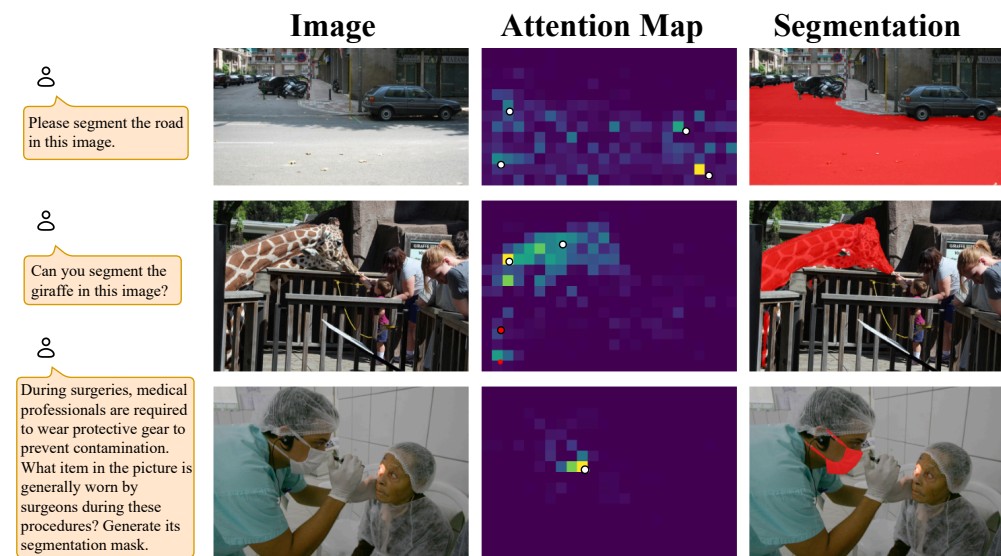

Figure 3: **Showcases of *LENS*.** Attention maps align well with ground-truth regions, where white points mark keypoints inside and red points mark those outside. These results illustrate how *LENS* links semantics with segmentation.

Table 1: **Comprehensive comparison.** *LENS* attains state-of-the-art segmentation with lower training cost while preserving general abilities. Training memory values with underlines mark inference overhead of the MLLM. *Seg* denotes segmentation data (semantic, referring, reasoning), *FP-Seg* augments it with false-premise samples, and *VQA* represents corpus from generative vision–language tasks. READ uses the largest training set, whereas *LENS* relies solely on *Seg*.

| Method | Backbone | Training Mem (GB)↓ | Training Data Seg | FP-Seg | VQA | Seg ↑ | MME | MMBench | MMMU | MMStar |
|---|---|---|---|---|---|---|---|---|---|---|
| random guess | – | – | – | – | – | – | 1050.0 | 25.0 | 26.8 | 24.6 |
| LLaVA-1.5-7B | – | – | | | ✓ | – | 1808.4 | 66.5 | 35.7 | 33.1 |
| SESAME | LLaVA-1.5-7B | 30 × 8 | | ✓ | ✓ | 30.4 | 1394.4 | 28.3 | 11.2 | 20.3 |
| LISA | LLaVA-1.5-7B | 30 × 8 | ✓ | | ✓ | 56.9 | 184.5 | 0 | 0 | 0 |
| READ | SESAME | 30 × 8 | ✓ | ✓ | ✓ | **58.6** | 476.3 | 0 | 1.1 | 14.4 |
| *LENS* (Ours) | LLaVA-1.5-7B | 16 + 10 × 8 | ✓ | | | 57.3 | **1801.4** | **64.0** | **34.4** | **33.3** |

gineering, yet its dual-objective paradigm inherently weakens both. Overall, *LENS* achieves state-of-the-art segmentation while fully preserving general capabilities. In contrast, prior approaches that train the MLLM inevitably suffer severe degradation, often performing worse than random guessing.

## 4.3 SEGMENTATION EVALUATION

We reported the segmentation performance of *LENS* on both reasoning segmentation (Table 2) and referring segmentation (Table 3). Fig. 3 qualitatively illustrated the progression from attention maps to extracted keypoints and the final segmentation masks.

**Strong Performance on Both Reasoning and Referring Segmentation.** As shown in Table 2 and Table 3, *LENS* achieves strong performance on both reasoning segmentation (ReasonSeg) and referring segmentation (RefCOCO(+/g)). On ReasonSeg, *LENS* reaches 65.3 cIoU on validation and 57.3 on test, outperforming LISA (62.9/56.9) under the same LLaVA-1.5-7B backbone. Its performance is also comparable to READ, even though READ benefits from SESAME-based initialization and substantially more training data. On RefCOCO(+/g), *LENS* achieves 70.3, exceeding LISA-7B (69.8) and markedly outperforming non-MLLM baselines such as LAVT (66.5) and CRIS (64.3).

Table 2: **Comparisons on the ReasonSeg dataset.** The best performance is highlighted in **bold**, and the second best is underlined.

| Method | val | | test | | | | | |
| | overall | | short query | | long query | | overall | |
| | gIoU | cIoU | gIoU | cIoU | gIoU | cIoU | gIoU | cIoU |
|---|---|---|---|---|---|---|---|---|
| SESAME (Wu et al., 2024b) | 34.8 | 39.1 | 28.3 | 27.6 | 31.6 | 32.7 | 30.5 | 30.4 |
| LLaVA + OVSeg (Liang et al., 2023) | 38.2 | 23.5 | 24.2 | 18.7 | 44.6 | 37.1 | 39.7 | 31.8 |
| LISA-7B (Lai et al., 2024) | 52.9 | 54.0 | 40.6 | 40.6 | 49.4 | 51.0 | 47.3 | 48.4 |
| LISA-LLaVA-1.5-7B (Lai et al., 2024) | 61.3 | 62.9 | 48.3 | 46.3 | 57.9 | 59.7 | 55.6 | 56.9 |
| READ-7B (Qian et al., 2025) | 59.8 | **67.6** | **52.6** | **49.5** | **60.4** | 61.0 | **58.5** | **58.6** |
| *LENS*-7B | **61.4** | 65.3 | 47.8 | 41.7 | 59.3 | 61.6 | 56.5 | 57.3 |
| $14^{th}$ layer $\rightarrow 30^{th}$ layer | 45.7 | 43.6 | 32.6 | 35.7 | 39.6 | 40.8 | 37.9 | 40.0 |
| w/o keypoint description | 51.8 | 48.5 | 42.1 | 39.3 | 49.8 | 49.8 | 47.9 | 47.8 |
| w/o global description | 56.0 | 61.9 | 44.0 | 40.3 | 50.4 | 49.4 | 48.8 | 47.9 |
| w/o $\mathcal{L}_{attn}$ | 55.8 | 51.7 | 42.9 | 40.5 | 54.6 | 53.4 | 51.7 | 50.8 |

Table 3: **Comparison of SOTA referring segmentation (cIoU) on RefCOCO(+/g).**

| Method | RefCOCO | | | RefCOCO+ | | | RefCOCOg | | Mean |
| | val | testA | testB | val | testA | testB | val(U) | test(U) | |
|---|---|---|---|---|---|---|---|---|---|
| MCN (Luo et al., 2020) | 62.4 | 64.2 | 59.7 | 50.6 | 55.0 | 44.7 | 49.2 | 49.4 | 54.4 |
| VLT (Ding et al., 2021) | 67.5 | 70.5 | 65.2 | 56.3 | 61.0 | 50.1 | 55.0 | 57.7 | 60.4 |
| CRIS (Wang et al., 2022) | 70.5 | 73.2 | 66.1 | 62.3 | 68.1 | 53.7 | 59.9 | 60.4 | 64.3 |
| LAVT (Yang et al., 2022) | 72.7 | 75.8 | 68.8 | 62.1 | 68.4 | 55.1 | 61.2 | 62.1 | 66.5 |
| ReLA (Liu et al., 2023a) | 73.8 | 76.5 | 70.2 | 66.0 | 71.0 | 57.7 | 65.0 | 66.0 | 68.3 |
| X-Decoder (Zou et al., 2023a) | – | – | – | – | – | – | 64.6 | – | – |
| SEEM (Zou et al., 2023b) | – | – | – | – | – | – | 65.7 | – | – |
| SESAME (Wu et al., 2024b) | 74.7 | – | – | 64.9 | – | – | 66.1 | – | – |
| LISA-7B (Lai et al., 2024) | 74.9 | **79.1** | **72.3** | 65.1 | 70.8 | 58.1 | 67.9 | **70.6** | 69.8 |
| *LENS*-7B | **76.5** | 78.3 | 71.4 | **66.1** | **71.7** | **58.3** | **69.4** | 70.6 | **70.3** |

**Improvement Room on Short Queries.** ReasonSeg training set is highly imbalanced, as it was originally designed for training MLLMs with explanatory content in combination with VQA data. Since *LENS* can only leverage the segmentation portion, which contains no short-query samples, performance on this category dataset remains limited.

**Ablation Study.** We evaluated the contributions of key components on the ReasonSeg dataset (*c.f.* Table 2), focusing on their effect on cIoU. The steepest drop occurs when shifting the head's input from the $14^{th}$ to the $30^{th}$ layer, which reduces cIoU from 57.3 to 40.0. This decline arises because features from deeper layers lose spatial detail and exhibit weak cross-modal attention, leaving the head unable to exploit the MLLM's intrinsic spatial cues. Likewise, removing the keypoint module or the attention loss lowers performance (47.8 and 50.8), underscoring the importance of explicit spatial signals. Finally, omitting the global description slightly affects validation performance but significantly harms test performance, highlighting its role in supporting generalization.

# 5 CONCLUSION

This work establishes *LENS* (**L**everaging k**E**ypoi**N**ts for MLLMs' **S**egmentation) as a plug-and-play architecture that brings segmentation into MLLMs without compromising their general-purpose abilities. By freezing the entire MLLM and introducing a lightweight head that leverages the model's own spatial cues as keypoints, *LENS* bypasses the objective conflict that hampers prior fine-tuning-based approaches. Our experiments demonstrate that *LENS* achieves state-of-the-art segmentation performance while preserving the MLLM's broad capabilities and cutting training costs by a large margin. These results highlight *LENS* as an efficient and scalable paradigm for extending MLLMs, marking a step toward unified vision models that combine high-level reasoning and ultimately encompass the full spectrum of visual tasks.

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

# Segmentation as a Plug-and-Play Capability for Frozen Multimodal LLMs

## Supplementary Material

In the supplementary materials, we report:

- Additional implementation details of our method, including keypoint sampling, sub-pixel optimization, and the specific structure of the descriptor (§S1);

- The integration of *LENS* into an agent system (§S2);

- Detailed experiment settings (§S3);

- More showcases of *LENS* (§S4);

## S1 ADDITIONAL IMPLEMENTATION DETAILS

### S1.1 KEYPOINT SAMPLING

Given the attention map $A_c^1$, we first reshape it into a 2D heatmap of size $h \times w \times 1$. To extract salient keypoints, we apply a non-maximum suppression (NMS) strategy on this heatmap. Specifically, we iteratively select the location with the highest response value as a keypoint, then suppress all responses within a fixed Euclidean radius $r = 4$ pixels around the selected location by setting their values to zero. This procedure is repeated until either no remaining responses exceed zero or the number of selected keypoints reaches a predefined upper limit $N = 16$. The resulting set of spatial coordinates corresponds to the most discriminative local regions in the attention map, ensuring a sparse yet informative representation while avoiding redundant neighboring points.

### S1.2 SUB-PIXEL REFINEMENT

Because the attention map is low resolution (*e.g.*, LLaVA-1.5-7B yields a $24 \times 24 \times 1$ heatmap), integer-coordinate keypoints may be spatially biased. We therefore refine each integer keypoint to sub-pixel precision by locally fitting a second-order Taylor model of the heatmap and taking a single Newton step.

**Setup.** Let the batched heatmaps be $H \in \mathbb{R}^{B \times K \times \hat{H} \times \hat{W}}$ and the corresponding integer keypoints be

$$\mathbf{p}_{b,k}^{\text{int}} = (x_{b,k}, y_{b,k}) \in \{0, \dots, \hat{W} - 1\} \times \{0, \dots, \hat{H} - 1\},$$

for batch index $b \in \{1, \dots, B\}$ and keypoint index $k \in \{1, \dots, K\}$. We define the normalized grid coordinates used for bilinear sampling (align_corners = true):

$$\tilde{x}_{b,k} = 2\,\frac{x_{b,k}}{\hat{W} - 1} - 1, \qquad\qquad \tilde{y}_{b,k} = 2\,\frac{y_{b,k}}{\hat{H} - 1} - 1, \qquad (1)$$

$$\Delta_x = \frac{2}{\hat{W} - 1}, \qquad\qquad \Delta_y = \frac{2}{\hat{H} - 1}. \qquad (2)$$

**Local sampling.** Around each integer keypoint we bilinearly sample the heatmap at the $3 \times 3$ neighborhood (center, 4-neighbors, and 4 diagonals) in normalized coordinates:

$$
\begin{aligned}
v^{(0)} &= H\big(\tilde{x},\ \tilde{y}\big), & v^{(1)} &= H\big(\tilde{x} + \Delta_x,\ \tilde{y}\big), & v^{(2)} &= H\big(\tilde{x} - \Delta_x,\ \tilde{y}\big), \\
v^{(3)} &= H\big(\tilde{x},\ \tilde{y} + \Delta_y\big), & v^{(4)} &= H\big(\tilde{x},\ \tilde{y} - \Delta_y\big), & v^{(5)} &= H\big(\tilde{x} + \Delta_x,\ \tilde{y} + \Delta_y\big), \\
v^{(6)} &= H\big(\tilde{x} - \Delta_x,\ \tilde{y} - \Delta_y\big), & v^{(7)} &= H\big(\tilde{x} - \Delta_x,\ \tilde{y} + \Delta_y\big), & v^{(8)} &= H\big(\tilde{x} + \Delta_x,\ \tilde{y} - \Delta_y\big).
\end{aligned}
$$

$$(3)$$

(For brevity we drop indices $b, k$ and the channel dimension; sampling is applied per $(b, k)$.)

**Finite-difference estimates of derivatives.** Using the samples in equation 3, we estimate first- and second-order partial derivatives at the center point via standard central differences:

$$D_x = \frac{1}{2}\big(v^{(1)} - v^{(2)}\big), \qquad\qquad D_y = \frac{1}{2}\big(v^{(3)} - v^{(4)}\big), \qquad (4)$$

$$D_{xx} = v^{(1)} - 2v^{(0)} + v^{(2)}, \qquad\qquad D_{yy} = v^{(3)} - 2v^{(0)} + v^{(4)}, \qquad (5)$$

$$D_{xy} = \frac{1}{4}\big(v^{(5)} + v^{(6)} - v^{(7)} - v^{(8)}\big). \qquad (6)$$

These define the local gradient $\mathbf{g} \in \mathbb{R}^2$ and Hessian $\mathbf{H} \in \mathbb{R}^{2\times 2}$:

$$\mathbf{g} = \begin{bmatrix} D_x \\ D_y \end{bmatrix}, \qquad \mathbf{H} = \begin{bmatrix} D_{xx} & D_{xy} \\ D_{xy} & D_{yy} \end{bmatrix}. \qquad (7)$$

**Regularized Newton step.** We obtain the sub-pixel offset $\boldsymbol{\delta} \in \mathbb{R}^2$ by a regularized Newton update of the quadratic model: $\boldsymbol{\delta} = -\big(\mathbf{H} + \varepsilon\mathbf{I}\big)^{-1}\mathbf{g}$ with a small Tikhonov term $\varepsilon > 0$ (e.g., $\varepsilon = 10^{-6}$) to improve numerical stability. To prevent spurious large corrections in flat or noisy regions, we clip the offset component-wise: $\boldsymbol{\delta} \leftarrow \mathrm{clip}(\boldsymbol{\delta}, -1, 1)$

**Refined coordinates (pixel space).** The refined sub-pixel keypoint in pixel coordinates is

$$\mathbf{p}_{b,k}^{\mathrm{sub}} = \mathbf{p}_{b,k}^{\mathrm{int}} + \boldsymbol{\delta} = \begin{bmatrix} x_{b,k} \\ y_{b,k} \end{bmatrix} + \begin{bmatrix} \delta_x \\ \delta_y \end{bmatrix}. \qquad (8)$$

If $\mathbf{H}$ is ill-conditioned, a diagonal fallback can be used: $\delta_x = -D_x/(D_{xx} + \varepsilon)$, $\delta_y = -D_y/(D_{yy} + \varepsilon)$.

### S1.3 Descriptor Model

We provide a detailed description of the descriptor model used in the second stage, as illustrated in Fig. S1. The inputs to this model are the global semantic feature $f_s^2$ and a set of local keypoint feature vectors $\{d_{\mathcal{N}(i)}\}$. Note that for each keypoint $i$, $d_{\mathcal{N}(i)}$ contains multiple feature vectors describing the surrounding local region. Therefore, we use $f_s^2$ as the *query* and all feature vectors within $d_{\mathcal{N}(i)}$ as the *keys* and *values*, and perform a cross-attention operation to jointly determine whether the corresponding region should be segmented. This process is repeated for all $m$ keypoints, yielding descriptors $\{d_i\}$. Concatenating them with the global feature $f_s^2$, we perform a self-attention refinement, followed by a projection to match the SAM decoder's dimension, producing the final set $D \in \mathbb{R}^{(m+1)\times d_s}$.

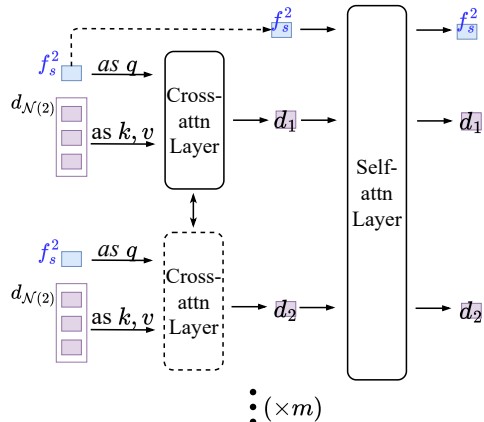

Figure S1: **The sturcture of the descriptor model.**

We interpret the meaning of $D$ as follows: each keypoint descriptor $d_i$ indicates whether its associated local region should be segmented, while the global descriptor $f_s^2$ provides holistic contextual information that coordinates and complements the local decisions across regions.

### S1.4 Training Strategy

Table 2 shows that removing either the global description or the keypoint description leads to a performance drop, indicating that both components contribute significantly to the overall performance. These two components need to be coordinated during training to achieve a better balance. To this end, we adopt a dropout-like mechanism: with a probability $p = 0.5$, we randomly use only

the global description or use both during training. During inference, both descriptions are always used. We find that this dropout mechanism generally ensures better performance. More fine-grained tuning of the balance between the two components may further boost performance, similar to how different settings of cIoU and gIoU affect performance.

## S2 AGENT INTEGRATION

Algorithm S1 illustrates how to integrate *LENS* into an agent framework (in a LangChain-style architecture) for multimodal interaction. Given a user instruction $u$ and an input image $I$, the agent $\mathcal{A}$ (the MLLM) first determines the *intent* of the instruction using a prompt-based classifier. Specifically, the instruction and image are given to $\mathcal{A}$ with a few-shot prompt that asks it to output one of the following three intent types:

- **Dialogue:** The instruction is a general conversational query unrelated to segmentation. In this case, the agent directly performs autoregressive generation conditioned on the text and image, and outputs a natural-language answer.

- **Segmentation:** The instruction explicitly asks to segment certain objects or regions in the image. The agent extracts an intermediate embedding from its internal representation and passes it to *LENS*'s head $\mathcal{H}$, which decodes a segmentation mask. The segmentation result is visualized and stored in the memory $\mathcal{M}$ for potential future reference, and the system returns a fixed textual response together with the visualized result.

- **Follow-up:** The instruction refers to the previously segmented content (*e.g.*, asking about the segmented object). The original image and its segmentation result are concatenated and passed back to the agent $\mathcal{A}$, which then answers the follow-up question based on both.

This design enables seamless switching between general dialogue and vision-centric segmentation tasks, while maintaining conversational context through memory.

---

**Algorithm S1:** Agent-guided Segmentation and Dialogue

**Input:** Instruction $u$, image $I$; Agent model $\mathcal{A}$; *LENS*'s head $\mathcal{H}$; memory $\mathcal{M}$

1  intent $\leftarrow \mathcal{A}$.Route$(u, I)$ ;                    // $\in \{\texttt{dialogue}, \texttt{seg}, \texttt{followup}\}$
2  **if** *intent = seg* **then**
3        // The user instruction contains a segmentation intent
3        $e \leftarrow \mathcal{A}$.Embed$(u, I)$;
4        mask $\leftarrow \mathcal{H}$.Decode$(e, I)$;
5        $\mathcal{M}$.last $\leftarrow (I, \text{mask})$;
6        **return** "Sure, the segmentation result is generated.", Overlay$(I, \text{mask})$;

7  **else if** *intent = followup* **then**
8        $(I_0, \text{mask}_0) \leftarrow \mathcal{M}$.last or $(I, \varnothing)$;
9        **if** $\text{mask}_0 = \varnothing$ **then**
10           **return** THISALGORITHM$(u, I)$;
11       $C \leftarrow \text{Concat}(I_0, \text{Overlay}(I_0, \text{mask}_0))$;
12       **return** $\mathcal{A}$.Generate$(u, C)$;

13 **else**
14       **return** $\mathcal{A}$.Generate$(u, I)$;

---

## S3 DETAILED SETTINGS

**Training Settings.** We clarify our training choices. Although one could follow READ by adopting a stronger backbone and incorporating broader *FP-Seg* data to obtain higher segmentation scores, such design diverges from our motivation. Our goal is not to improve segmentation accuracy by incremental modifications, but to explore a new architecture that preserves the general capabilities of MLLMs and advances toward a unified vision model. Using an MLLM already trained for segmentation as the backbone would contradict this objective, while *FP-Seg* introduces excessive generative

samples that are misaligned with our single segmentation objective. Therefore, we strictly follow the LISA training setup (but excluding VQA data).

**Evaluation Settings** We evaluate how *LENS* preserves the general capabilities of the underlying MLLM. As shown in Table 1, we introduce a `random guess` baseline to estimate the expected performance when answers are generated completely at random, since these benchmarks adopt multiple-choice formats.

*MME Benchmark.* The MME benchmark consists of 10 perception and 4 cognition subtasks (14 in total). Each image is paired with two binary (yes/no) questions. The official evaluation computes, for each subtask, the **accuracy** (fraction of correctly answered questions) and **accuracy+** (fraction of images where *both* questions are correct). The subtask score is defined as

$$\text{score} = 100 \times (\text{accuracy} + \text{accuracy+}).$$

Under random guessing ($p = 0.5$):

$$\mathbb{E}[\text{accuracy}] = 0.5, \quad \mathbb{E}[\text{accuracy+}] = 0.25, \quad \Rightarrow \ 100 \times (0.5 + 0.25) = 75.$$

Since there are 14 subtasks, the overall expected score is $14 \times 75 = 1050$.

*MMBench Benchmark.* We use the English test split of MMBench, which contains about 6.7K multiple-choice questions. Each question has four options with a single correct answer, and evaluation is conducted using overall **accuracy**. Under random guessing, the expected accuracy is 25% due to the $1/4$ selection probability.

*MMMU Benchmark.* MMMU contains about 11.5K multimodal questions from college-level exams and textbooks, spanning six broad disciplines, 30 subjects, and 183 subfields. It combines both multiple-choice and open-ended formats with highly diverse image types (charts, diagrams, maps, chemical structures, etc.). Following the official protocol, we evaluate on the public **validation split** containing 900 samples, and report overall **accuracy**. The expected random-guessing performance is provided in the official report.

*MMStar Benchmark.* MMStar is a vision-indispensable benchmark of 1,500 carefully curated samples covering six core capabilities and eighteen fine-grained axes. All questions are cast into a multiple-choice format, and we follow the official setting to report **accuracy** as the primary metric. Random-choice performance is provided by the official report and serves as the baseline reference.

Models performing below the `random-guess` baseline (such as LISA and READ) tend to ignore the question content and directly output segmentation-related responses, indicating that their general-purpose reasoning ability has been severely impaired. SESAME observed this issue and introduced additional false-premise data during training to mitigate it, but its performance remains only slightly above random guessing, further validating our hypothesis that dual-objective training damages general capability.

In contrast, *LENS* is specifically designed to avoid this issue by introducing segmentation capability in a decoupled, plug-and-play manner: it attaches an external head while keeping all MLLM parameters frozen, allowing the model to autonomously decide whether to invoke the segmentation head.

## S4    SHOWCASES

We present a comprehensive demonstration of *LENS*'s performance on both standard segmentation tasks and reasoning-based segmentation tasks in Figs. S2–S4. Typically, when the attention maps focus on correct regions and the keypoints are accurately localized (Fig. S2), the segmentation results are satisfactory. Even when some keypoints are mistakenly detected, the description mechanism in *LENS* can designate them as negatives and still produce correct segmentation results (Fig. S3). However, if the attention is largely distributed over non-target regions (Fig. S4), *LENS* may fail, resulting in incorrect segmentation.

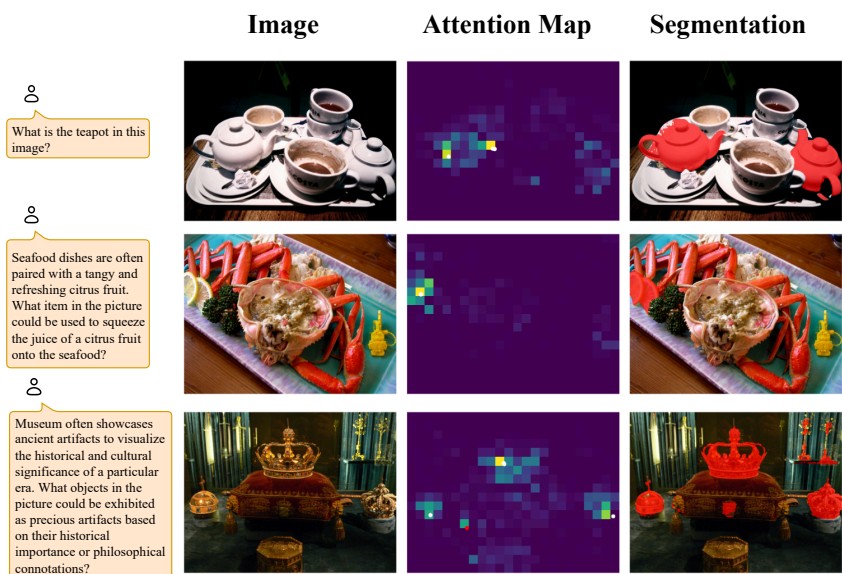

Figure S2: **Showcases of *LENS*.** The white dots overlaid on the attention maps indicate keypoints that are aligned with the target segmentation regions.

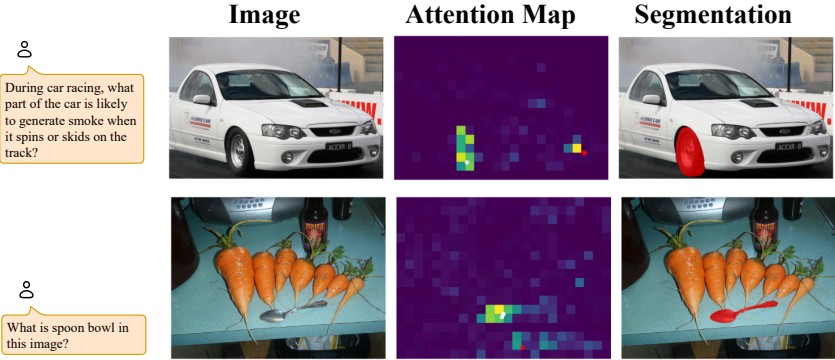

Figure S3: **Showcases of *LENS*.** The red dots on the attention maps denote keypoints located in non-target regions. Even when such keypoints are detected, the description mechanism in *LENS* ensures that the final segmentation results remain correct.

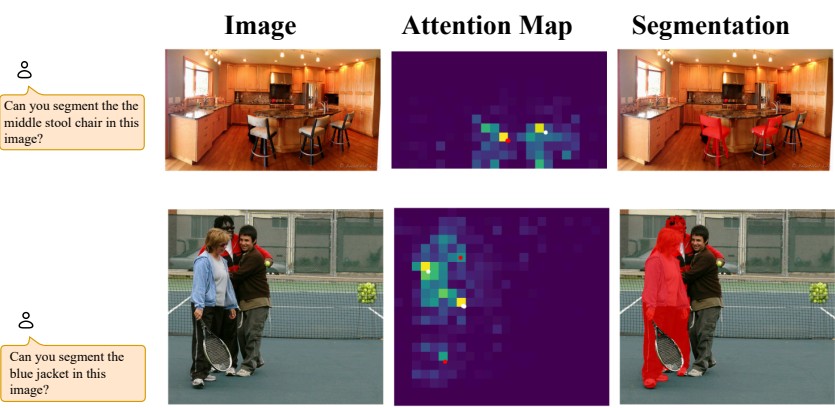

Figure S4: **Failure cases of *LENS*.** If the attention map shows strong responses on non-target regions, incorrect segmentation may occur.

