# OpenReview forum: "Segmentation as a Plug-and-Play Capability for Frozen Multimodal LLMs"
_ICLR.cc/2026/Conference — ICLR 2026 Conference Withdrawn Submission_

### Official Review · Reviewer_BX6Z · 2025-10-27

**Soundness:** 2
**Presentation:** 2
**Contribution:** 2
**Rating:** 4
**Confidence:** 3

**Summary:**

This paper presents LENS, a plug-and-play framework for enabling semantic segmentation in fully frozen MLLMs. The authors observe that intermediate-layer attention maps in MLLMs contain latent spatial cues, allowing pixel-level segmentation without joint fine-tuning. A lightweight two-layer Transformer segmentation head is attached on top of the frozen model to refine attention, extract keypoints, and decode masks using SAM. Experiments on datasets such as RefCOCO and ReasonSeg show that LENS achieves segmentation performance comparable to fine-tuned methods while preserving the original visual understanding capability.

**Strengths:**

1. **A novel paradigm.**
   The paper challenges the prevailing assumption that pixel-level capability requires joint fine-tuning. It demonstrates that a frozen MLLM can acquire segmentation ability through an external plug-in, which is conceptually significant for unified multimodal modeling.

2. **Clear and well-motivated architecture.**
   The two-layer segmentation head has distinct roles: Layer 1 refines spatial attention, while Layer 2 enhances semantic consistency. This modular design supports independent optimization and practical deployment.

3. **Comprehensive experiments.**
   The method is evaluated across semantic, referring, and reasoning segmentation benchmarks (COCO-Stuff, ADE20K, RefCOCO, ReasonSeg) with thorough ablations on keypoint description, layer selection, and attention loss, providing solid empirical support for the main claims.

**Weaknesses:**

1. **Insufficient motivation for the `training-free` and `non-cooperative` design.**
   While the paper emphasizes freezing the MLLM and training only an external head, it does not clearly articulate the practical or theoretical value of such a non-cooperative framework. The results show feasibility, but the rationale for avoiding joint optimization remains underexplained.

2. **Unconvincing argument for the segmentation–generalization trade-off.**
   The paper treats the drop in generalization performance after segmentation fine-tuning as an inherent conflict between tasks, without distinguishing it from common out-of-distribution degradation. Treating performance loss from domain-specific fine-tuning as evidence of an intrinsic conflict is not fully convincing.

3. **Unclear relation to more general MLLMs.**
   Recent unified models such as BAHEL[1] and Lumina-mGPT[2] can perform both image understanding and pixel-level editing, which are arguably more complex than pure segmentation. LENS’s novelty lies mainly in its frozen and modular design, and its independent value relative to these unified MLLMs should be better clarified.

4. **Weak empirical validation of semantic alignment.**
   Layer 2 is claimed to align the [s] token with image features, yet there is no quantitative or visual evidence supporting this assumption.

> [1]: Emerging Properties in Unified Multimodal Pretraining

> [2]: Lumina-mGPT: Illuminate Flexible Photorealistic Text-to-Image Generation with Multimodal Generative Pretraining

**Questions:**

1. How do the authors interpret the true significance of “training-free” segmentation? Is freezing the backbone a principled choice or merely a pragmatic one?
2. If limited fine-tuning were allowed, would LENS still preserve both visual understanding and segmentation performance? Can the degree of conflict between these two tasks be quantified?
3. Please address the issues listed in the Weaknesses section, especially the attention-scale bias and semantic alignment validation.
4. In Equation (1), because of the causal mask, attention from later text tokens to image tokens may be diluted by earlier tokens, leading to an overall decrease in aggregated attention energy. It is not clear whether the authors have considered the potential impact of this phenomenon on the computation of $A^1_c$.
5. See `weakness`.

---

### Official Review · Reviewer_fR3S · 2025-10-27

**Soundness:** 3
**Presentation:** 3
**Contribution:** 2
**Rating:** 4
**Confidence:** 4

**Summary:**

This paper introduces LENS (Leveraging keypoints for MLLMS' Segmentation) as a plug to enhance segmentation ability for modern MLLMs. LENS attaches a two-layer trainable head to a frozen MLLM and extracts and refines keypoints as soft prompts in SAM based on the attention map between user's text query and the input image. The proposed plugin is trained on different segmentation datasets and validated across various benchmarks.

**Strengths:**

- The proposed plugin is able to maintain the general ability of the foundation MLLMs. LENS achieves segmentation by adding two extra learnable layers to extract and refine keypoint features. The added two layers are light-weighted and easy to train as it is not distracted by other MLLM tasks. The original ability on general multimodal QA tasks is also maitained by utiliting a routing network as the original head in LLava keeps intact.

- The ablation studies are conducted in an overall organized manner. I appreciate the ablation over "w/o keypoint description", which is equivalent to a baseline that only adopts the extracted "raw keypoints" and feeds them into SAM. This solves the reader's concern on the necessity of the keypoint description module in LENS

- The evaluations are considered comprehensive with general abilities including MME, MMBench, MMMU, and MMStar and also segmentation abilities including RefCOCO, RefCOCO+, RefCOCO+, and ReasonSeg datasets.

**Weaknesses:**

- This work leverages the aggregated attention between the text tokens and image tokens as localization and segmentation cues. Similar ideas have been actively applied to image segmentation recently[1-3]. Some other references are also discussed in the related work of this paper as well. The authors are encouraged to highlight the uniqueness of this paper.

- This paper claims maintaining the general ability of the original MLLM by introducing another head for segmentation. However, the ability is a result of a routing network that decides whether to call the original or the new head. The routing design is also applicable to other segmentation methods as well. Therefore, it is recommended to set a new baseline that also uses the routing network so that the contribution of this work is better differentiated.

- The routing network is introduced to identify the users intention. However, the accuracy is the routing network is not discussed in the manuscript. The end-to-end cIoU performance including both the LENS segmentation and the accuracy of the routing network is recommended to be reported.

- The baselines that this method compares with could be more comprehensive and up-to-date. Some of recent work that enhance the segmentation ability of MLLMs are not covered and compared to in the manuscript. Here list a few work with performance on RefCOCO-val enclosed in brackets that adopt the same LLM structure with LENS:  Text4Seg (77.5), POPEN (79.3), SegLLM (80.2), GLaMM (79.5)
These works achieves higher cIoU than LENS (76.5) but not rigorously discussed in the paper.

- Apart from the above literature adopting the same LLM structure, there is a trend that uses a stronger MLLM (e.g. Qwen-VL) to further the state-of-art performance. I would suggest the authors conducting a similar study to show its generalization across different structures and scaling effect, which could create better influence as well.

**Questions:**

Some minor questions:
- To what extent do the hyperparameters of NMS influence the performance?

The reviewer would also like to hear from the authors about the rebuttal on the uniqueness issue and a fair comparison in order to readjust the rating score.

---

### Official Review · Reviewer_iRxk · 2025-10-29

**Soundness:** 3
**Presentation:** 3
**Contribution:** 2
**Rating:** 2
**Confidence:** 5

**Summary:**

This paper introduces LENS, a novel "plug-and-play" architecture to add segmentation capabilities to MLLMs. The key idea is to attach a lightweight, trainable head to a completely frozen MLLM backbone. This head refines the MLLM's internal attention maps to extract spatial cues, identifies keypoints from these maps, and generates descriptions for these keypoints. These keypoint-description pairs are then used as prompts for a frozen SAM-based decoder to produce the final segmentation mask.
The primary motivation is to solve the "inherent tension" in previous methods (e.g., LISA) that fine-tune MLLMs with dual objectives. The authors argue this dual training degrades the MLLM's foundational general-purpose abilities

**Strengths:**

- The paper's strongest contribution is its solution to the "objective conflict". By freezing the MLLM backbone and only training a lightweight head, the method preserves the MLLM's general-purpose capabilities while still enabling segmentation. This is the main advantage over prior methods that require extensive fine-tuning of the entire MLLM.
- The insight to repurpose existing internal representations (attention maps) rather than forcing the model to generate new ones (like a special [SEG] token ) is well-motivated.
- By keeping the MLLM frozen, it drastically reduces training memory requirements and avoids the computational cost of backpropagation through the main model.

**Weaknesses:**

- The method has lots of "handcrafted tricks" that undermine the "plug-and-play" claim. The design relies on fragile, non-trivial choices that are not easily generalized:
    - Specific layer choice: The method's success is critically dependent on using an intermediate layer (the 14th). The ablation study shows that moving to a deeper layer (the 30th) causes performance to collapse, demonstrating a lack of architectural robustness.
    - The design depends on a cascade of specific mechanisms, such as sub-pixel refinement, and using the "start-of-answer" token's feature as the semantic query. These choices seem tightly coupled to the LLaVA architecture and would likely require significant re-engineering to "plug" into other MLLMs.

- Limited novelty.
   - The proposed work is quite similar to [f], which also freeze the backbone and use attention map to get the final masks.

- Lack of comparison with SOTA and performance is not good.
    - Outdated baselines: The comparisons, especially in Table 3 for referring segmentation, are outdated. Baselines like LISA (2024) are not the current SOTA.
    - Missing key comparisons: The paper omits comparisons to numerous, highly relevant, and more recent (2024-2025) methods that directly address MLLM-based segmentation and vision-language tasks. The absence of methods like [a] - [e] is a major flaw. These works are the true SOTA, and without comparing against them, the paper's performance claims are unsubstantiated.

### **Conclusion**
While the paper presents a well-written and good story, its core contributions are undermined by some significant issues:
1. The method's "plug-and-play" claim is weak. The design is highly handcrafted and fragile, making it unlikely to be easily adapted to other MLLMs.
1. The experimental evaluation is incomplete. A lack of comparison to numerous, closely related SOTA methods hinders the credibility of the paper's performance claims.
1. The method's segmentation performance is substantially weaker than current SOTA. Its primary defense, the preservation of general abilities, is also not a unique contribution, as other methods [f] have already achieved this. This combination of inferior performance and a non-unique core benefit leaves the paper in an awkward position: the design appears elegant but with limited practical value.

[a] GLaMM: Pixel Grounding Large Multimodal Model. CVPR2024.
[b] SAM4MLLM: Enhance Multi-Modal Large Language Model for Referring Expression Segmentation. ECCV2024.
[c] Text4Seg: Reimagining Image Segmentation as Text Generation. ECCV2024.
[d] SegLLM: Multi-round Reasoning Segmentation with Large Language Models. ICLR2025.
[e] SegAgent: Exploring Pixel Understanding Capabilities in MLLMs by Imitating Human Annotator Trajectories. CVPR2025.
[f] F-LMM: Grounding Frozen Large Multimodal Models. CVPR2025.

**Questions:**

Please see weakness points above.

---

### Official Review · Reviewer_eiy1 · 2025-10-30

**Soundness:** 3
**Presentation:** 4
**Contribution:** 3
**Rating:** 6
**Confidence:** 3

**Summary:**

This paper proposes LENS, a plug-and-play module that equips MLLMs with segmentation capability. Unlike prior approaches that jointly finetune MLLMs with both segmentation and generation objectives, LENS keeps the MLLM frozen and only leverages its internal spatial cues derived from cross-modal attention. These cues are converted into keypoint prompts and fed into SAM for mask prediction, allowing the model to use the MLLM's reasoning ability without altering its output space. This design avoids the performance degradation and loss of generality commonly seen in fine-tuned approaches. Experiments show that LENS achieves comparable or better segmentation results than fine-tuning-based methods, while fully preserving the MLLM's general vision-language capabilities.

**Strengths:**

- **Clear plug-and-play paradigm.** The paper cleanly decouples segmentation from MLLM training and frames segmentation as an attachable capability rather than a jointly-learned objective, avoiding interference with the model's core abilities.

- **Insight on cross-modal attention as spatial signal.** The work is grounded on the observation that MLLM cross-modal attention already encodes localization cues, and operationalizes this insight instead of retraining the backbone to emit segmentation-specific features.

- **Well-aligned modular pipeline.** The design is structured and task-aligned: a lightweight head to refine attention and enhance semantics, keypoint extraction and description to bridge to SAM, and a dedicated mask decoder. Each module has a clear role and consistent data interface.

- **Strong performance without compromising generality.** The method achieves segmentation performance comparable to state-of-the-art fine-tuned approaches while preserving the MLLM's general vision-language capabilities, addressing the capability collapse seen in prior work.

- **Efficiency gains.** The frozen-backbone design significantly reduces memory and training cost, demonstrating practical advantages for scaling and deployment.

- **Clear presentation.** The paper is clearly written, with intuitive figures and a coherent flow that makes the core idea and pipeline easy to follow.

**Weaknesses:**

- While the keypoint-based prompting mechanism is elegant and efficient, it remains unclear whether a relatively small number of keypoints can reliably capture more complex or fine-grained object structures, especially in multi-object or highly detailed segmentation scenarios. Additional analysis or visualization on such cases could further strengthen the argument.

- The experiments primarily focus on LLaVA-1.5-7B as the backbone. Although the approach is conceptually model-agnostic, demonstrating results across diverse MLLMs would reinforce claims about general applicability and plug-and-play extensibility.

- The method supervises cross-modal attention via BCE against ground-truth masks. While empirically effective, attention and pixel-level masks are conceptually different signals; a short discussion on this alignment assumption and whether softer or alternative supervisory formulations could be explored would provide additional clarity.

**Questions:**

1. Some of my questions relate to the discussion points mentioned in the Weakness section.

2. In Table 2, could the ablation settings be described in a bit more detail? For instance, I am not entirely sure what "removing the keypoint module" exactly entails.

3. In Table 2, some rows highlight the best score but do not mark the second-best result. Is this intentional (e.g., only best values are emphasized), or simply a formatting choice?

---

### Note · Authors · 2025-11-21

I have read and agree with the venue's withdrawal policy on behalf of myself and my co-authors.